# Blueberry Anthocyanins from Commercial Products: Structure Identification and Potential for Diabetic Retinopathy Amelioration

**DOI:** 10.3390/molecules27217475

**Published:** 2022-11-02

**Authors:** Rui Li, Zhan Ye, Wei Yang, Yong-Jiang Xu, Chin-Ping Tan, Yuanfa Liu

**Affiliations:** 1School of Food Science and Technology, Jiangnan University, Wuxi 214122, China; 2State Key Laboratory of Food Science and Technology, Jiangnan University, Wuxi 214122, China; 3National Engineering Research Center for Functional Food, Jiangnan University, Wuxi 214122, China; 4Collaborative Innovation Center of Food Safety and Quality Control in Jiangsu Province, Wuxi 214122, China; 5Department of Food Technology, Faculty of Food Science and Technology, Universiti Putra Malaysia, Serdang 43400, Malaysia

**Keywords:** blueberry anthocyanins, cyanidin-3-o-glucoside, diabetic retinopathy, REDD1, molecular docking

## Abstract

The aim of the present study was to determine the major anthocyanins of blueberry extracts from northeast China and explore their vision health improvement effects. HPLC-Q-TOF-MS/MS results suggested that six different anthocyanins were accurately identified, among which the Cy-3-glu (C3G) was the most abundant, ranging from 376.91 ± 7.91 to 763.70 ± 4.99 μM. The blueberry extract contained a higher purity of anthocyanins, and the anthocyanosides reached 342.98 mg/kg. The anti-oxidative stress function of C3G on HG-treated ARPE-19 cells were evaluated, and showed that the GSSG level of HG-cells pretreated with 10 μM C3G was significantly decreased, while the Nrf2 and NQO1 gene expression levels were increased. Further molecular docking (MD) results indicated that the C3G displayed favorable binding affinity towards REDD1, and only the B-ring of the C3G molecule displayed binding interactions with the CYS-140 amino acids within the REDD1 protein. It implied that the oxidative stress amelioration effects of C3G on the ARPE-19 cells were related to the REDD1 protein, which was probably via the Nrf2 pathways, although further studies are needed to provide mechanism evidence. The present study provides novel insights into understanding the roles of blueberry anthocyanins in ameliorating oxidative stress-induced BRB damage in the retina.

## 1. Introduction

Vision health problems are one of the most important public health threats worldwide. The International Diabetes Federation (IDF) reported that the global prevalence of diabetes retinopathy (DR) was as high as 27% during the period 2015 to 2019, and 12.5% in Southeast Asia [1]. DR has become one of the leading vision health problems, and more attention must be paid to DR prevention.

Blueberry belongs to the genus *Vaccinium* and subgenus *Cyanococcus*, and is known as a ‘super fruit’; it is considered to have outstanding commercial value due to the high antioxidant capacity of abundant polyphenolic compounds [2,3]. They are rich in flavones and other phenolic compounds, and can be used as the raw materials for manufacturing functional foods. In recent years, they have attracted much attention for exhibiting a broad array of biological functions, such as antioxidant and anti-inflammatory capabilities, preventive neurodegeneration and cardiovascular disease properties, and the amelioration of vision effects, etc. [4,5,6]. The anthocyanins are the most important group of bioactive compounds in plants, including blueberries and bilberries, and polyphenols of the flavonoid group [7]. Many previous studies demonstrated that high doses of anthocyanins showed potential for prevention or treatment of type 2 diabetes [8,9]. Whereas, anthocyanins played more important roles in DR improvement than their hypoglycemic effect. For instance, in a double-blind, monocentric, randomized, and prospective study, Macuprev^®^ supplement (containing anthocyanosides 90 mg) increased the function of the macular pre-ganglionic elements, which possibly contributed to reduce inflammation in DR lesions [10]. Therefore, anthocyanins, due to their strong anti-oxidation effects, might be an alternative relief strategy in the early stage of DR progression, apart from the medical interventions.

Extensive studies have been conducted to explain the mechanism of DR progression [11]. They have shown that the vascular endothelial growth factor (VEGF) plays an important role in vasculogenesis and angiogenesis, and affects the blood retinal barrier (BRB) under DR pathological conditions [12]. Clinical research has shown an important relationship between BRB and retinopathy; consequently, many animal studies and in vitro studies mainly focused on the effects of anthocyanins on BRB. For example, bilberry extracts displayed outstanding effects in the prevention of diabetes-induced retinal vascular dysfunction in a diabetic rat model, and the content of the total anthocyanins under the optimal dose for the vision BRB breakage inhibition was estimated as 36 mg/kg [13]. 

Although a previous study showed that the blueberry anthocyanin extract and its predominant constituent (malvidin-3-glucoside) can decrease the angiogenesis of the cell model induced with DR by decreasing the VEGF level and inhibiting the Akt pathway [4], these results did not indicate that blueberry extracts or anthocyanin monomer compound showed effects on relieving DR. Most probably, the anti-VEGF therapy mainly targeted the microvascular defects and neovascularization that occur in the advanced stage of DR, and there was evidence that the VEGF level remained low and relatively unchanged compared with nondiabetic controls in the early stage of the DR [14]. Therefore, it is necessary to find new targets to clarify the effect of anthocyanins in DR. 

Regulated in development and DNA damage 1 (REDD1), a stress response protein, was required for upregulating translation of the mRNA encoding VEGF, whose expression was increased in diabetes-induced visual dysfunction [15]. Moreover, it showed that REDD1 could characterize the initial deficits of neuro-glial in the early hyperglycemia phases of diabetes [16]. According to the research results, REDD1 was not only required for VEGF overexpression induced by diabetes, but also contributed to inflammatory signaling related to DR. Therefore, REDD1 might be a target of DR treatment [16,17]. Notably, REDD1 expression was positively correlated with the overproduction of reactive oxygen species (ROS) induced by high-glucose in cell studies [17]. It provides a theoretical basis for studying the blueberry extracts or anthocyanin monomer compounds effect on BRB through regulation of REDD1. 

Therefore, in the present study, firstly, the chemical structures of anthocyanins from different blueberry extracts, which were obtained from Northeast China, the main region of blueberry planting, were identified by high performance liquid chromatography coupled with tandem mass spectrometry (HPLC-Q-TOF-MS/MS), and their content were qualified by high performance liquid chromatography coupled with a diode array detector (HPLC-DAD). Secondly, the antioxidant activity of the different blueberry extracts was evaluated by Ferric-reducing antioxidant power (FRAP) and DPPH radical scavenging capacity (DPPH), followed by GSSG and GSH assay for evaluation of the effect of the monomer compound cyanidin-3-O-glucoside (C3G) treatment on the ARPE-19 cell redox environment. Thirdly, the possible role of the different anthocyanins from blueberry extracts in REDD1 production inhibition was explored by using molecular docking technique. The overall purpose of the present study was to provide theoretical guidance for the anthocyanin identification of the commercial blueberry extract products; furthermore, we lay the foundations for DR prevention by using anthocyanins-enriched dietary supplements.

## 2. Results and Discussion

### 2.1. Identification of Major Anthocyanin Compounds from the Blueberry Extracts

The phenolic compounds from blueberry extracts are typically divided into flavonoids and non-flavonoids. The anthocyanins are a class of natural bioactive substances that contain a wide variety of subspecies and belong to the flavonoid categories [2]. Different compounds can be detected by an HPLC detector under a specially designed detection wavelength; for instance, the accurate qualitative and quantitative analysis of flavan-3-ols, flavanols, and anthocyanins could be realized at 280, 360, and 520 nm by HPLC with a photo-diode-array-detector, respectively [18]. Presently, we firstly used the HPLC-DAD for the isolation of the total anthocyanin compounds, which were subsequently identified by the HPLC-Q-TOF-MS/MS for uncovering their detailed chemical structures.

As shown in Figure 1A, anthocyanins detected at 520 nm were the major phenolic compounds in the S1 sample from Northeast China, and the retention time was 8–14 min (Figure 1A(b)). Additionally, the S1 sample contained a small amount of flavan-3-ols and flavanols, which were detected at 280 and 360 nm, respectively (Figure 1A(a)). By comparing different blueberry extract samples, the results showed that all four of the other samples showed similar phenolic compound profiles as the S1 sample, in which the anthocyanins were dominant, while a small number of phenolic acids and flavone glycosides were also found in the S2–S5 samples (Appendix A).

Therefore, the isolated primary anthocyanins at 520 nm were identified by MS, and the results are shown in Figure 1B–G and Table 1. Peak 3 was identified by external experiments with the same authentic standards as cyanidin-3-O-glucoside (C3G) (RT = 10.55 min) (Appendix A). According to the above results, C3G was the most abundant anthocyanin in the anthocyanin-rich extract from the five samples. Peaks 2, 3, and 5 showed the [M+] ion with m/z values of 449.1027, 449.1085, and 419.0899, respectively. In the MS/MS product ion scan, the anthocyanins contained common sub-ions with an m/z of 287.055, which indicated that they included the basic parent structure of cyanidin. Galactose, glucose, and arabinose are common aglycones linked to the parent structure, and the retention time from short to long was galactose, glucose, and arabinose [19]. According to their retention time and documentary evidence, these compounds were positively identified as cyanidin-3-O-galactoside, cyanidin-3-O-glucoside, and cyanidin-3-O-arabinoside. Peak 4 showed the [M+] ion *m/z* value of 595.1651, which suggested that it was disaccharide derivatives of anthocyanins. The MS/MS product ion scan produced a fragment ion with an *m*/*z* of 287.055, corresponding to loss of a hexose and a pentose molecule attached at the same position. The most common pentose glycosides in nature are arabinose and xylose, with the former usually preferred [20]. Thus, on the basis of the UV−vis profile and MS data, peak 4 was proposed to be cyanidin-3-O-rutinoside. Peak 6 showed an ion with an m/z of 433.1183 and fragment ion with an *m*/*z* of 271.0473, coincident with formula C_21_H_21_O_10_, which was identified as pelargonidin-3-O-hexose. All four compounds had been identified in blueberries (*Vaccinium uliginosum* L.) and bilberries (*Vaccinium myrtillus* L.) [19]. For peaks 6, the hexose was probably the epimers galactose or glucose, each presenting different polarities as shown by the difference in retention time and structure. Lacking sufficient information, the specific identity could not be determined. In a similar manner, Peak 7 showing an ion with an *m*/*z* of 463.1223 and fragment ion with an *m*/*z* of 301.0634 was identified as peonidin-3-O-hexose; it was previously reported in wild Chinese blueberries [21]. 

These results were consistent with those of previous studies characterizing the detailed compositions of the phenolic compounds from the blueberry, and it was also suggested that the flavonoid polyphenolic compounds were mainly anthocyanins (up to 60%), followed by procyanidins and flavanols [3]. Notably, the anthocyanin content of sample S1 from Northeast China seems to be much higher than that of anthocyanins reported in other studies. Subsequently, the contents of anthocyanins of different blueberry extracts were accurately quantified to provide valuable information for explaining the function of anthocyanins in performing redox balance regulation. 

### 2.2. Quantitative Analysis of the Total Phenols and Anthocyanins Contents of Different Blueberry Extracts

Anthocyanins are a class of phenolic compounds. In in order to identify the specific proportion of anthocyanins in blueberry extracts, the content of total phenols and anthocyanins was determined, respectively. The total phenolic content (TPC) of the five different blueberry extracts samples was quantitatively measured by using the Folin-Ciocalteu reagent, and their concentrations were expressed as micrometer (μM) gallic acid equivalents (GAE). The content of the total anthocyanins, as well as the individual anthocyanins from different blueberry extract samples, were analyzed by HPLC-DAD, and the results were expressed as C3G equivalents (C3GE) (μM), according to the European Pharmacopoeia principle [22]. 

The results showed that the TPC of the studied blueberry extract samples ranged from 758.03 ± 27.11 to 878.10 ± 35.13 μM GAE (Figure 2A). The TPC of the S4 sample was found to be the highest (878.10 ± 35.13 μM GAE), and the S5 sample was the lowest (758.03 ± 27.11 μM GAE). Furthermore, no significant difference was shown between the S1 sample sourced from Northeast China (838.38 ± 3.25 μM GAE) and other samples. As shown in Figure 1, the five samples contained similar anthocyanosides, which were Cy-3-gal, Cy-3-glu, Cy-3-rut, Cy-3-ara, Pg-3-hex, and Pn-3-hex; and the six anthocyanosides were mainly Cy-based anthocyanins. However, the anthocyanin contents of five blueberry extracts were different. As demonstrated in Figure 2B, the total anthocyanins content (TAC) of the different samples ranged from 376.91 ± 7.91 to 763.70 ± 4.99 μM C3GE. The types of anthocyanins in blueberries reported in the literature were different because there were high variations among blueberry cultivars for total anthocyanin content and their species, which were reported mainly due to their genotypes and growing conditions [23]. Although only 6 major anthocyanins were identified in this study, the content of total anthocyanins had obvious advantages. In particular, the S1 sample contained the highest level of anthocyanosides (763.70 ± 4.99 μM C3GE), which was equivalent to 342.98 mg/kg in anthocyanins. This was much higher than the content of anthocyanins used in many previous large-scale clinical studies (25 or 36 mg/kg in anthocyanins) [10]. Studies suggested that highly purified anthocyanin-rich extracts showed higher potential in terms of antioxidants [24], which might be of benefit in the amelioration of the vision function disorders related with oxidative stress [25]. Furthermore, the C3G was the dominant anthocyanins monomer among the others (201.61−548.36 μM), and the C3G content of the S1 sample was significantly higher compared with that of the other samples, about twice that of the other blueberry extract samples. This might imply that the S1 sample could be used as an alternative natural bioactive ingredient to ameliorate oxidative stress related to retinal disorders.

Collectively, it was demonstrated that the blueberry extract from Northeast China contained relatively higher content of anthocyanins compared with the others. The bioavailability of the anthocyanins had been considered an important problem for the utilization of this bioactive product, which might lead to their poor functionality for relieving oxidative stress-related disorders [26]. However, previous studies suggested that the anthocyanins showed appreciable biological functions in protecting RPE cells from oxidative stress-induced damage when they were treated at very high dosages of individual anthocyanins [27,28]. Therefore, the antioxidant properties of the individual anthocyanins might deserve to be re-evaluated, which is also important for illustrating their bio-functional roles for improving host health.

### 2.3. Antioxidant Activities of Five Blueberry Extracts and the Anthocyanin Monomer C3G

The antioxidant capacity of anthocyanins is generally relatively lower than that of other phenolic compounds, such as flavanols and phenolic acids [29]. This lower capacity can be unfavorable when evaluating the antioxidant activity of the S1 sample. In order to explore the antioxidant activity of high-purity anthocyanin extracts in vitro, the Ferric-reducing antioxidant power (FRAP) and DPPH radical scavenging capacity of the five different blueberry extract samples were assessed, and the results were expressed as gallic acid equivalent (GAE) [30]. Simultaneously, we also used the same methods to measure the antioxidant activity of the anthocyanin monomer C3G for reference.

As shown in Figure 3A, the results from the FRAP and DPPH assays showed similar trends. Specifically, the antioxidant capacity of sample S1, which was qualified by the FRAP assay, was slightly but significantly lower than that of the other four samples (*p* < 0.05); moreover, no significant difference could be observed between the S2, S3, S4 or S5 samples (Figure 3A(a)). The DPPH scavenging abilities of S1 showed a significant difference with S2 and S4 (*p* < 0.05), but no significant difference with S3 and S5 (Figure 3A(b)). Overall, sample 1 had poor antioxidant capacity compared with the other four samples, which might be due to the fact that it contained less phenolic acids and flavone glycosides compared with other samples. However, this did not mean that the S1 sample had no advantage in antioxidant capacity. One study showed that the antioxidant capacity of blueberries was mainly owing to the contained anthocyanins, which accounted for as much as 84% of the total antioxidant capacity [31]. Figure 3A shows that the antioxidant capacity of sample 1 was only slightly different from that of the other samples. 

Finally, the antioxidant capacity of the most abundant monomer compound C3G detected in the different samples was evaluated, and it showed that the DPPH scavenging ability was 80.58 ± 1.95 μM GAE and FRAP was 359.92 ± 32.62 μM GAE (Appendix A), confirming that the monomer compound C3G displayed stronger antioxidant capacity. One study showed that C3G reduced retinal degeneration induced by diabetes by scavenging ROS [32]. Therefore, we speculated that anthocyanins may protect retinal barrier cells from damage by reducing ROS.

### 2.4. Effect of C3G Treatment on ARPE-19 Cells Redox Environment

This section analyzes the redox status of ARPE-19 cells pre-treated with C3G before the high glucose (HG) culture to further determine the biological function of the most abundant anthocyanin monomer C3G in blueberry extract. First, the content of the glutathione disulfide-glutathione couple (GSSG/GSH), an important index for evaluating the cellular redox environment, was measured. Next, the mRNA expression levels related to oxidative stress were determined considering the effect of C3G on gene level.

As shown in Figure 3B, the intracellular GSSG level was significantly increased (*p* < 0.05) from 0.308 to 0.507 μM after HG treatment, while the GSH level was decreased (*p* = 0.092), indicating weakened cellular glutathione metabolism after HG induction. However, anthocyanin pretreatment restored the disordered glutathione metabolism. Compared with the HG model groups, the GSSG level of HG-cells treated with 10 μM C3G was significantly decreased (*p* < 0.05) and the GSH levels were also increased by about 30%, which was recovered to the levels in the blank control, suggesting that the GSH signaling was involved during this recovering process. Furthermore, the ARPE-19 cells treated with 10 μM of C3G alone displayed a significant increase in the GSH level (*p* < 0.001) and a significant decrease in the GSSG level (*p* < 0.01), which indicated that the 10 μM C3G intervention could perform cellular oxidative stress recovery functions without leading to cytotoxicity. 

The Nrf2 was the central control factor of the expression of its downstream oxidases NQO1 under normal homeostasis and oxidative stress [33]. Therefore, the expression levels of Nrf2 and NQO1 was measured on a transcriptional level for confirming the effects of C3G for alleviating the HG-induced cellular oxidation stress (Figure 3C). The mRNA expression levels of both Nrf2 and NQO1 in ARPE-19 cells were significantly decreased in response to the 30 mM HG exposure for 24 h (*p* < 0.001). Whereas, the Nrf2 and NQO1 mRNA expression levels were increased after C3G pretreatment, and these were inconsistent with the GSSG and GSH results. The results showed that HG-induced ARPE-19 cellular oxidative damage could be effectively restored, and the redox balance was also improved. 

Because the oxidative stress played an important role in the pathogenesis and pathophysiology of DR development, intervention by natural phenolic compounds for regulating the cellular redox balance might be a new proposed relief strategy for DR [34]. Furthermore, researchers have well documented that Nrf2 activation by anthocyanins can upregulate expression levels of antioxidant enzymes and proteins, such as glutaredoxin1 (Grx1), heme oxygenase-1 (HO-1), thioredoxin 1 (Trx1), and NADPH dehydrogenase 1 (NQO1) [35]; and lower blood sugar via the Nrf2/Keap1 pathway [36]. The retinal pigment epithelium (RPE) is the outer barrier of the retina, which could protect the retina from oxidative stress-induced impairment. The above results indicated that C3G protected the barrier cells (ARPE-19) from oxidative stress. Another study further confirmed that ROS level and endoplasmic reticulum stress in ARPE-19 cells induced by HG was notably restored by blueberry extract, and the development of DR in diabetes rat models was alleviated [37].

Collectively, these data provide evidence that blueberry extracts, which are rich in C3G, might protect ARPE-19 cells against HG-induced oxidative stress via the Nrf2 pathway, although the underlying mechanisms need to be explored on the gene translation level. 

### 2.5. Molecular Docking between Anthocyanin Monomer Compounds and REDD1

Numerous previous studies have suggested that REDD1 was not only a positive regulator of cellular reactive oxygen species (ROS), but also a negative regulator of antioxidant response [17,38]. Diabetes can lead to an increase in ROS production and antioxidant defense system impairment, and these changes are closely connected with the endogenous oxidative stress of the local microenvironment, for example, in the local retina. One study found that REDD1 could maintain a high ΔΨ m via suppression of Akt/GSK3β signaling [17]. GSK3β also directly phosphorylated the transcription factor Nrf2 to promote its nuclear exclusion and attenuated expression of antioxidant genes with antioxidant response element (ARE) promoters [39]. These indicated that the REDD1 sensed the changes in Nrf2, which were interfered with by the turbulence of oxidation stress, which meant Nrf2 and REDD1 share the same signal pathway and can translate the pathological condition associated with the vision disorders. These mechanism signal paths play key roles in DR development and progression, and have been widely explored in many previous studies [39]; however, recent studies investigating the roles of anthocyanins for vision health promotion, or eyesight protection mainly focused on the Nrf2 rather than the REDD1. A study showed that baicalein, as a natural flavone, caused a marked induction of DDIT4, which also called REDD1 that mediated mTOR inhibition and growth inhibition in cancer cells [40]. Therefore, we speculated that anthocyanins might be closely related to the production of REDD1.

If there are connections between the anthocyanins and REDD1, and whether the anthocyanins can directly act on the REDD1 protein is still uncovered. Therefore, to provide a more in-depth understanding about the roles of anthocyanins for vision health improvement in view of the REED1 protein, molecular docking between the different anthocyanins and REDD1 protein was analyzed. Five different anthocyanidins were docked into the catalytic site of REDD1. The predicted binding modes of five main anthocyanidins, including Cy-3-glu, Cy-3-rut, Cy-3-ara, Pg-3-glu, and Pn-3-glu docked into REDD1 are shown in Figure 4A–E, respectively. 

Figure 4 shows that REDD1 encapsulated anthocyanins in the structural cavity, and the five anthocyanin molecules docked with the amino acid residues of REDD1 through the hydrogen bond. According to the conformational analysis of the binding action between REDD1 and anthocyanins, the binding affinities of the five anthocyanidins were presented in Table 2. The results showed that Pg-3-glu exhibited the strongest binding affinity of REDD1 (−7.69 kcal/mol), followed by Cy-3-glu (−7.54 kcal/mol, C3G); and Cy-3-rut was the worst (−6.21 kcal/mol).

According to previous studies, the antioxidant capacity of anthocyanins was related to the number of hydroxyl groups in the B-ring, and the hydroxyl groups on the carbon site 4 of the B-ring are the most active groups [41]. The number of H-bonds of C3G and Pg-3-glu was found to be 10 and 6, respectively; however, only the B-ring of the C3G molecule was linked to the hydroxyl groups with CYS-140 on REDD1, indicating that C3G might be the optimal anthocyanin for inhibiting REDD1 activity, and further perform BRB protection effects under oxidative stress. Additionally, the anthocyanins consist of anthocyanidins and aglycons, and the activity of anthocyanins largely depends on the anthocyanidins and their extent of glycosylation. Liu et al. reported that the different glycosides within the anthocyanins affected the chemical structure of anthocyanins; for example, the addition of disaccharide to the pure anthocyanin molecule affected the polarity of anthocyanins, and these might enhance or decrease the activity of anthocyanins [42]. Therefore, Cy-3-rut might not be as effective as C3G to act on REDD1, and the lower binding energy (−6.21 kcal/mol) between Cy-3-rut and REDD1 might be related with connected disaccharide. Furthermore, C3G was the most dominant anthocyanin in blueberry extracts, and its binding affinity toward REDD1 was relatively higher than the others. Thus, it can be concluded that C3G might be used as an alternative natural bioactive ingredient for amelioration of retinal oxidative stress by interacting with REDD1. However, more studies should be conducted in the future to verify these mechanisms using modern molecular analysis methods.

## 3. Materials and Methods

### 3.1. Materials and Reagents

The blueberries came from local varieties in the Great Khingan Mountains (China) and were harvested in 2021. Blueberry extracts were obtained from Daxinganling Lingonberry Boreal Biotech Co., Ltd. (Qiqihaerz, Heilongjiang, China), using the ethanol extraction method, acquired by multiple extraction. The samples were frozen immediately after picking and stored at −20 °C until use. Cyanidin-3-O-glucoside standard (>98%) was purchased from Standard Biotechnology Co., Ltd. (Shanghai, China). 2,4,6-tri(2-pyridyl)-s-triazine (TPTZ), 2,2-diphenyl-1-picrylhydrazyl (DPPH), methanol (HPLC grade), and formic acid (reagent grade) were purchased from Sigma-Aldrich (St. Louis, MO, USA). Folin-Ciocalteu (reagent grade) was supplied by Sinopharm Chemical Reagent. All other chemicals were of analytical or chromatographic grade and purchased from Sinopharm Chemical Reagent Co. Ltd. (Shanghai, China) and Thermo Fisher Scientific (Waltham, MA, USA). Milli-Q water (Milli-Q Direct 8, Millipore, Burlington, MA, USA) was used in the present study. 

### 3.2. Quantification of the Total Phenols and Anthocyanins Content of the Blueberry Samples

#### 3.2.1. Standard Solutions Preparation

Stock solutions of the cyanidin-3-O-glucoside (C3G) standard and five blueberry extracts were prepared in acidified water (2% HCl) at concentrations of 1 mM. These solutions were stored in brown bottles and kept at −20 °C in darkness. Working standard solutions were freshly prepared by diluting a certain volume of stock standard solutions with MeOH according to the concentration demanded before analysis. 

#### 3.2.2. Total Phenolic Content Quantification

The total phenolic content (TPC) of the different blueberry extracts was determined using Folin-Ciocalteu reagent, following the method of Singleton and Rossi with slight modification [43]. In brief, 1 mL of blueberry extract was mixed with 1 mL of Folin-Ciocalteu reagent followed by the addition of 5 mL of distilled water in a volumetric flask. Then, 3 mL of sodium carbonate (7.5% *w*/*v*) was added. After 5 min of vortex, the reaction was undertaken at room temperature for 120 min. The absorbance values at 750 nm were detected using a UV-VIS spectrophotometer. The total phenolic content of the samples was calculated from the calibration curve of standard gallic acid (0–1.5 mM) and their concentrations were expressed as gallic acid equivalents (Y = 0.0013x + 0.0902, R^2^ = 0.9981) (Appendix A).

#### 3.2.3. Anthocyanins Component Quantification

The HPLC system was equipped with an UltiMate3000 diode array detector (Thermo Fisher Scientific, Massachusetts, American). Chromatographic separations were conducted on a 150 mm × 4.6 mm, 5 μm Athena C18-WB column (ANPEL Laboratory Technologies, Shanghai). The mobile phase consisted of two solvents: formic acid (3%, *v*/*v*) in water (A) and methanol (B) with gradient elution at a flow rate of 1 mL/min. The gradient program of the mobile phase was as follows: 0–2 min, 6% B; 2–5 min, 6–10% B; 5–10 min, 10–16% B; 10–16 min, 16% B; 16–22 min, 16–24% B; 22–26 min, 24–32% B; 26–30 min, 32–40% B; 30–32 min, 40–90% B; 32–34 min, 90% B; 34–39 min, 90–0% B; and 39–40 min, 0% B. The injection volume was 10 μL for extracts of blueberries, and the column temperature was maintained at 40 °C. Specifically, spectral data from all peaks were accumulated in the range 190–600 nm, and chromatograms were recorded at 280 nm for flavan-3-ols, at 360 nm for flavanols and at 520 nm for anthocyanins. For quantification anthocyanins, their concentrations were expressed as C3G equivalents (C3GE) (μM), and the calibration curve of Y = 0.4924x − 5.0662 (R^2^ = 0.9991) (Appendix A).

### 3.3. The Structure Identification of the Different Anthocyanins from Blueberry Samples by HPLC-Q-TOF-MS/MS

A Waters ACQUITY high-performance LC system equipped with a Waters Xevo TQS mass detector was employed for anthocyanins analysis [44]. The separation chromatographic column was a reversed-phase Hypersil GOLD C18 (100 mm × 2.1 mm, 1.9 μm; Thermo Scientific, Waltham, MA, USA). The mobile phase consisted of two solvents: formic acid (0.5%, *v*/*v*) in water (A) and methanol (B) with gradient elution at a flow rate of 0.3 mL/min. The gradient program of the mobile phase was as follows: 0–2 min, 5% B; 2–5 min, 5–10% B; 5–10 min, 10–16% B; 10–12 min, 16–95% B; 12–15 min, 95% B; 15–16 min, 95–5% B; 16–20 min, 5% B. The injection volume was 3 μL, and the column temperature was maintained at 40 °C. 

The mass spectrometer was operated in a positive-ion mode by scanning ions between *m/z* 100 and 1000. The ESI inlet conditions were as follows: capillary voltage, 0.6 kV; cone voltage, 30 V; extractor voltage, 2 V; source temperature, 120 °C; and desolvation temperature, 350 °C. Data acquisition of anthocyanins and further determinations were done by the OS 1.8 software (Waters).

### 3.4. Evaluation of Antioxidant Capacity In Vitro

#### 3.4.1. DPPH Assay

The antioxidant capacity of five blueberry extracts was evaluated by 2,2-diphenyl-1-picrylhydrazyl (DPPH) radical scavenging capacity assay and ferric-reducing antioxidant power (FRAP) assay, respectively. The DPPH assay was performed according to the method reported in the previous studies with some modifications [29]. In brief, 270 μL of freshly prepared DPPH radical (80 μM in methanol) was mixed with 30 μL of blueberry extracts (200 μM in methanol). The reaction was conducted in a 96-well microplate at 25 °C. The absorbance values at 515 nm were detected at 30 min of the reaction by a plate reader (Model: SpectraMax190, Molecular Devices Inc., San Jose, CA, USA). The absorbance values of gallic acid in methanol ranging from 10 to 100 μM were used to construct the calibration curves. The antioxidant activity was expressed as gallic acid equivalent (μM). 

#### 3.4.2. FRAP Assay

The FRAP assay was performed using the method described by Yang, et al. [30]. The freshly prepared FRAP reagent (300 mM acetate buffer at pH 3.6, 10 mM TPTZ in 40 mM HCl, 20 mM FeCl_3_·6H_2_O in H_2_O, 10:1:1, *v*/*v*/*v*) was diluted to one-third with acetate buffer (300 mM acetate buffer at pH 3.6) and incubated at 37 °C. A total of 270 μL of diluted FRAP reagent was mixed with 30 μL of blueberry extracts (200 μM in methanol) in a 96-well microplate (Model: SpectraMax190, Molecular Devices Inc., USA). The reaction was undertaken at 37 °C for 4 min, and the absorbance values were measured at 593 nm against the blank, which consisted of 270 μL of diluted FRAP reagent and 30 μL of methanol. The absorbance values of gallic acid in methanol ranging from 20 to 200 μM were used to construct the calibration curves. The reduction ability was expressed as gallic acid equivalents (μM).

### 3.5. Redox Environment Evaluation of the ARPE-19 Cells Treated with Anthocyanin Monomer C3G

ARPE-19 cells (National Collection of Authenticated Cell Cultures, Shanghai, China) were cultured in Dulbecco’s modified Eagle’s medium (Sigma, USA) with 10% FBS, and the cell culture protocols were referred from Ma, et al. [45]. To examine the effects of C3G on a high glucose-induced cellular redox state, the cells were cultured in Minimum Essential Medium (MEM) without FBS for 12 h, then pretreated with C3G (10 μM) for 1 h, followed by 30 mM glucose treatment for another 24 h. The cells were used for GSSG/GSH assay, in order to evaluate the cellular redox environment by using the GSH and GSSG Assay Kit (Beyotime Biotechnology, Shanghai, China), respectively, according to the manufacturer’s protocols. Normal glucose (5.5 glucose) groups without C3G pretreatment were used as the control. The high glucose (30 mM glucose) groups without C3G pretreatment were used as the oxidative model. 

### 3.6. The Expression Levels of Genes Related with Redox Balance by Quantitative Real-Time PCR (RT-qPCR)

MolePure^®^ cell total RNA kit was used for total RNA preparation from ARPE-19 cells. The total RNA was used for cDNA preparation by using the Haifair^®^ III 1st Strand cDNA Synthesis SuperMix for qPCR (gDNA digester plus). The cDNA preparation was conducted following the manufacturer’s protocols. RT-qPCR was performed on an appliedbiosystems thermocycler (QuantStudio 3, Thermofisher, Waltham, MA, USA) using Hieff^®^ qPCR SYBR^®^ Green Master Mix (No Rox). The target primers were obtained from PrimerBank, which are displayed in Appendix A. The cycling conditions were as follows: the initial denaturation at 95 °C for 5 min, then 60 cycles at 95 °C for 10 s, plus 55–60 °C for 20 s, and finally at 72 °C for 20 s, which was performed for RT-PCR parameters: The GAPDH gene was used as an internal control to normalize target gene expression. The relative gene-expression value was calculated based on the 2 ^−ΔΔCT^ method, in which the ARPE-19 cells treated with 5.5 mM glucose was set as the control, and each sample was run in triplet. 

### 3.7. Molecular Docking Analysis between the REDD1 Protein and Different Anthocyanin Monomers

In order to determine the interactions between the stress response protein regulated in development and DNA damage 1 (REDD1) and anthocyanins, molecular docking analysis was conducted. The structure of the REDD1 (PDB code: 3LQ9) was downloaded from the PDB Protein Data Bank (https://www.uniprot.org/), accessed on 23 March 2022. For the optimization of the protein, the removal of the water molecules, and the addition of all the hydrogen atoms, the AutoDock Tools was used. A 3D structure of Cyanidin-3-O-glucoside (Cy-3-glu, CID: 197081), Cyanidin-3-O-rutinoside (Cy-3-rut, CID: 441674), Cyanidin-3-O-arabinoside (Cy-3-ara, CID: 12137509), Pelargonidin-3-O-glucoside (Pg-3-glu, CID: 3080714) and Peonidin-3-O-glucoside (Pn-3-glu, CID: 443654) were obtained from the National Centre for Biotechnology Information (NCBI) PubChem compound database and employed for ligand docking. The optimal docking results were obtained and further analyzed based on the principle of minimum binding energy and the maximum number of junctions in the operation results. The details of the interactions were displayed by using Pymol 3.7.7 (Python Software Foundation, Wilmington, DE, USA).

### 3.8. Statistical Analysis

The data was expressed as means ± standard deviation. Differences between groups were statistically analyzed using one-way analysis of variance (ANOVA) followed by Tukey’s post hoc test using SPSS (Version 22.0, SPSS Inc., Chicago, IL, USA) and considered statistically significant at *p* < 0.05 (*) and *p* < 0.01 (**). Pearson’s R correlation was also calculated using SPSS. The GraphPad Prism 9.0 (USA, GraphPad Software) was used for graphic design in the present work. 

## 4. Conclusions

The structure of the typical blueberry anthocyanins from commercial products was identified and their potential for diabetic retinopathy amelioration was explored by using a molecular docking technique. The results suggested that all the samples were rich in anthocyanins, and six major anthocyanosides were isolated and identified by HPLC-Q-TOF-MS/MS, among which the C3G was dominant and accounted for more than 50% of the total anthocyanins in different samples. Notably, the monomer compound C3G within the blueberry extracts showed high antioxidant capacity, and the C3G might perform oxidative stress amelioration effects through the Nrf2 pathways. Further molecular docking studies demonstrated that the C3G exhibited good binding affinity toward REDD1 (−7.54 kcal/mol) among the others, and only the B-ring within the C3G molecule displayed favorable binding interactions with the CYS-140 amino acids of the REDD1 protein, implying that the oxidative stress amelioration effects of C3G on HG-treated ARPE-19 cells was also connected with the REDD1 protein. The present study might provide some new insights into understanding the roles of blueberry anthocyanins for ameliorating oxidative stress-induced BRB damage in the retina. However, the underlying molecular mechanisms of natural anthocyanins in ameliorating the BRB injuries by regulating the REDD1 expression, and the relationships between Nrf2 and REDD1 still need to be further explored.

## Figures and Tables

**Figure 1 molecules-27-07475-f001:**
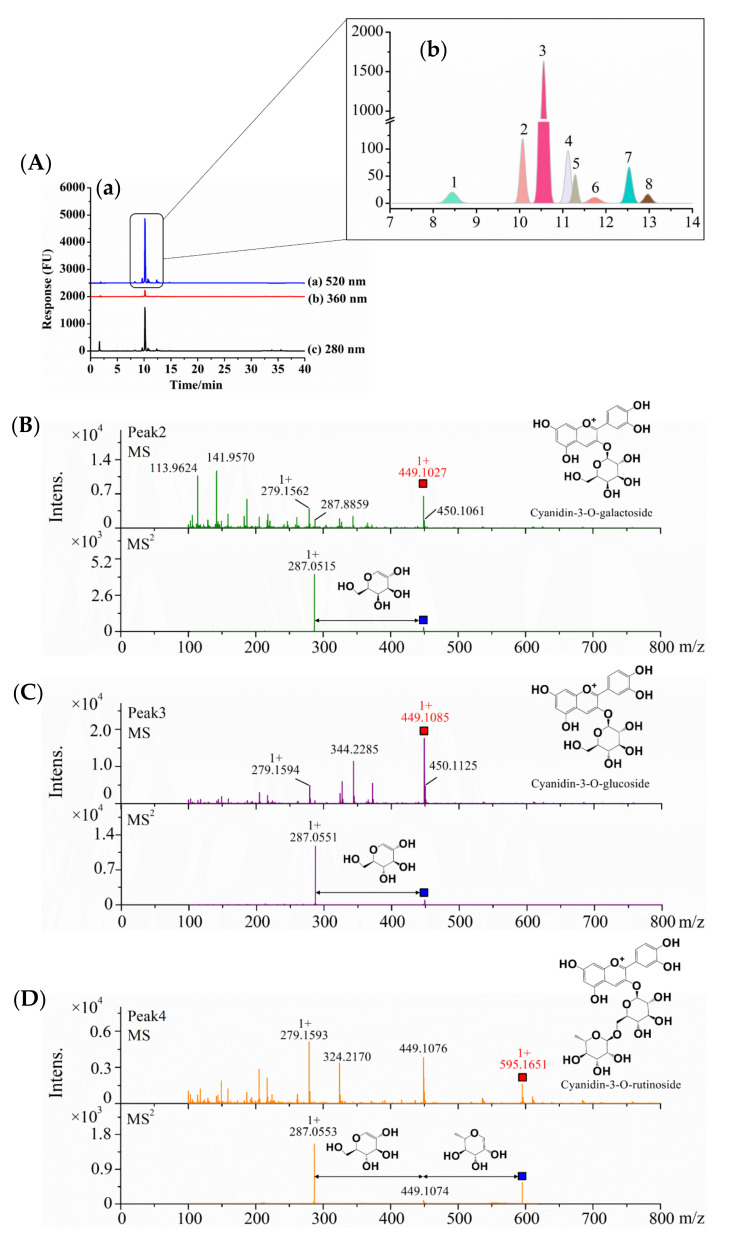
Identification and analysis of anthocyanins in five blueberry extracts. (**A**) HPLC chromatogram with UV detection at 280, 360, and 520 nm of S1 from Northeast China. (**B**–**G**) +MS and +MS2 of six different skeletons of anthocyanin compounds in five blueberry extracts under positive ion mode [M+] by HPLC-MS/MS spectra.

**Figure 2 molecules-27-07475-f002:**
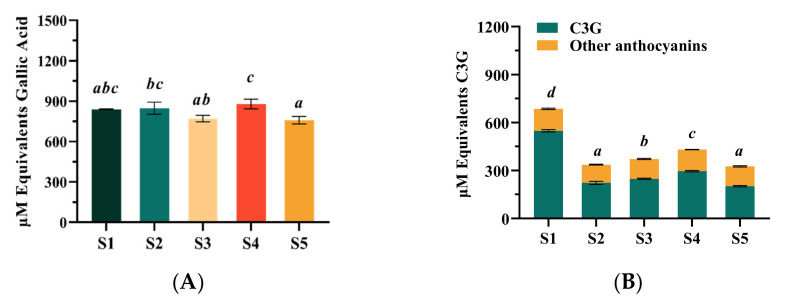
Concentration of anthocyanins and total phenol content (TPC) from five blueberry extracts. (**A**) TPC expressed as gallic acid equivalents (μM); (**B**) total anthocyanins content expressed as cyanidin-3-O-glucoside equivalents (μM). Different letters were significantly different (*p* < 0.05).

**Figure 3 molecules-27-07475-f003:**
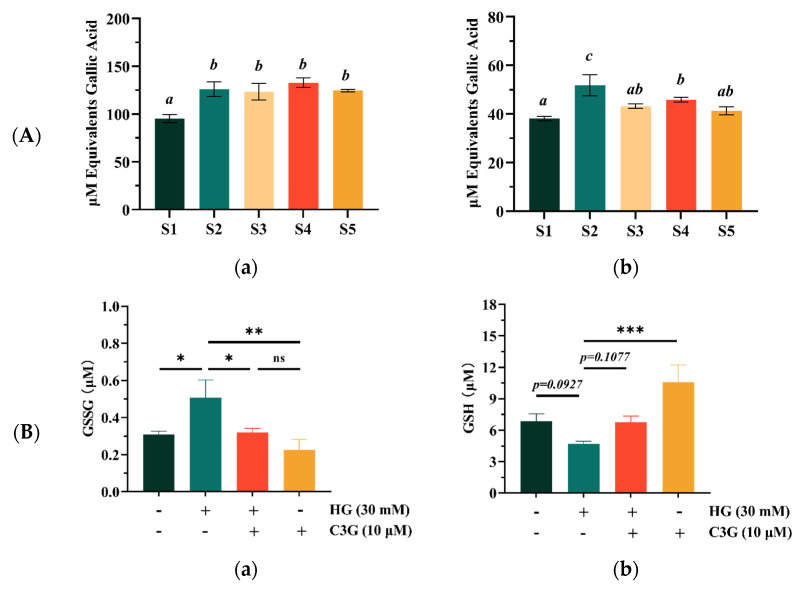
The antioxidant activities of the different blueberry extracts samples and monomer compound cyanidin-3-o-glucoside (C3G). (**A**) The Ferric-reducing antioxidant power (FRAP) (**a**) and DPPH radical scavenging capacity (DPPH) (**b**) of the different blueberry extracts samples; the concentration of blueberry extracts samples was 200 μM; (**B**) Effects of C3G treatment on GSSG (**a**) and GSH levels (**b**) in the ARPE-19 cells; (**C**) Effects of C3G treatment on the gene-expression of Nrf2 (**a**) and NQO1 (**b**) in the ARPE-19 cells. Data represent the mean ± standard deviation (*n* = 3). Different letters were significantly different (*p* < 0.05). (*) *p* ≤ 0.05, (**) *p* ≤ 0.01, and (***) *p* ≤ 0.001 compared with the high glucose (HG) only group.

**Figure 4 molecules-27-07475-f004:**
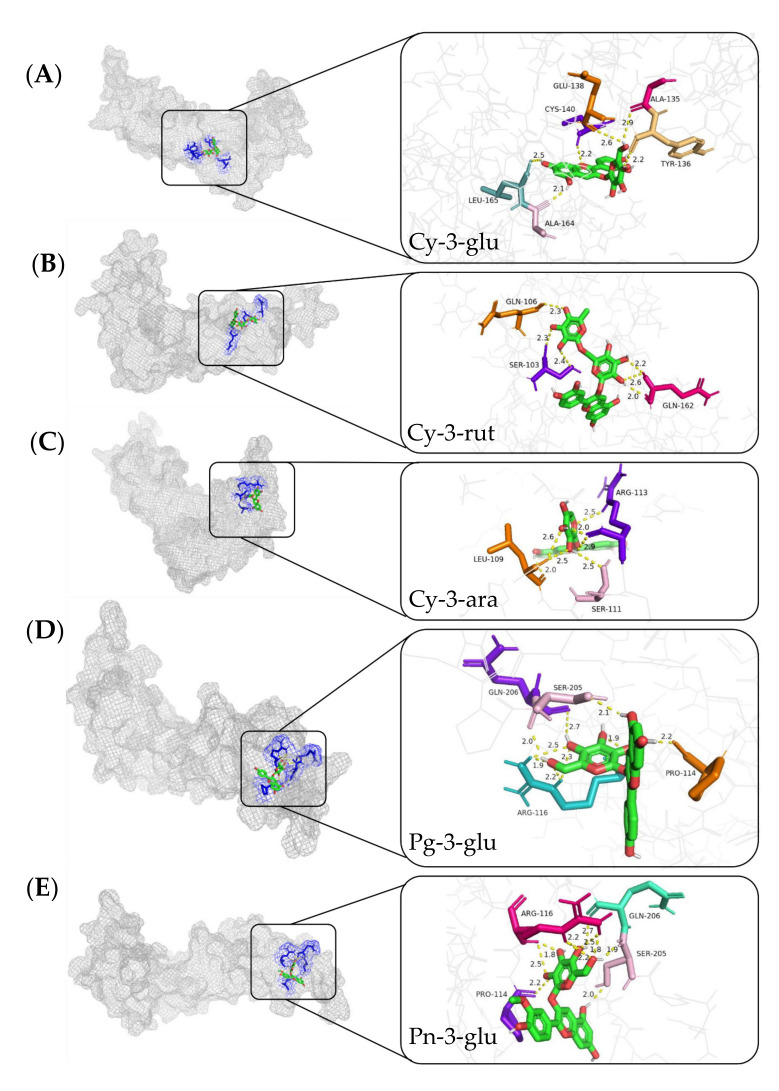
Use of Autodock software to analyze 100 docking models of five anthocyanin monomer compounds and proteins. Panoramic view showing the binding mode between Cy-3-glu (**A**), Cy-3-rut (**B**), Cy-3-ara (**C**), Pg-3-glu (**D**), Pn-3-glu (**E**), and REDD1.

**Table 1 molecules-27-07475-t001:** Identification of anthocyanins in five blueberry extracts and peak numbers as in Figure 1A(b).

Peak No.	RT (min)	MW	MS (*m/z*)	MS^2^ (*m/z*)	Aglycon	Sugar Moiety	Formula	Identification *
2	10.073	449	449	287	Cy	Hexose	C_21_H_21_O_11_	Cy-3-gal
3	10.547	449	449	287	Cy	Hexose	C_21_H_21_O_11_	Cy-3-glu
4	11.123	595	595	287, 449	Cy	Hexose + deoxyhexose	C_27_H_31_O_15_	Cy-3-rut
5	11.273	419	419	287	Cy	Pentose	C_20_H_19_O_10_	Cy-3-ara
6	11.727	433	433	271	Pg	Hexose	C_21_H_21_O_10_	Pg-3-hex
7	12.54	463	463	301	Pn	Hexose	C_22_H_23_O_11_	Pn-3-hex

* Note: Cy, cyanidin; Pg, pelargonidin; Pn, peonidin; gal, galactoside; glu, glucoside; rut, rutinoside; ara, aranoside.

**Table 2 molecules-27-07475-t002:** Predicted binding affinity for anthocyanidins present in extracts docked with REDD1.

Name	Docking Score (kcal/mol)	Numbers of H-Bonds	Amino Acid Residue
Cy-3-glu	−7.54	6	ALA-164, LEU-165, CYS-140, GLU-138, ALA-135 and TYR-136
Cy-3-rut	−6.21	6	GLN-106, SER-103 and GLN-162
Cy-3-ara	−7.30	6	LEU-109, SER-111 and ARG-113
Pg-3-glu	−7.69	10	GLN-206, SER-205, ARG-116 and PRO-114
Pn-3-glu	−6.92	10	GLN-206, SER-205, ARG-116 and PRO-114

## Data Availability

The data presented in this study are available on request from the corresponding author. The data are not publicly available due to privacy restrictions.

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
