# Peer review of "Blueberry Anthocyanins from Commercial Products: Structure Identification and Potential for Diabetic Retinopathy Amelioration"

_molecules, 2022, doi:10.3390/molecules27217475_

Round 1

Reviewer 1 Report

The manuscript entitles as “Structure identification of the typical blueberry anthocyanins from marketed products and exploring their potential for diabetic retinopathy amelioration” presented well. The manuscript can be accepted after following revisions.

1)    These anthocyanins have been identified previously with more number, although you have identified only 6, You must elaborate and highlight the importance of your study in both abstract and Result & Discussion portions.

2)    The antioxidant activity seems a bit confusing, explain in the section 2.3, the comparison of DPPH and FRAP, as both are quite different from each other.

3)    Insert the following reference as both related to Q-TOF-MS/MS if you find it useful. “10.3389/fphar.2022.890649” and “10.1016/j.jff.2021.104602”

4)    Check the whole manuscript for English as, correct the spelling, table S1 “aequences”. First sentence of third paragraph in introduction, line 58, remove the repeating word. Correct spelling Figure S4, “FRAR”.

Author Response

Thank you very much for your valuable comments and suggestions. From your comments and suggestions, I know you are definitely an expert in the research fields related with the present manuscript. To follow the suggestions, we have carefully revised the previous manuscript, and a point-by-point response to your comments, as well as the list of changes according to those comments and suggestions are listed as below:

  1. Q1: These anthocyanins have been identified previously with more number, although you have identified only 6, You must elaborate and highlight the importance of your study in both abstract and Result & Discussion portions.

Response: Thank you very much for your comments and suggestions. The significance of this research can be highlighted by adding the discussion content of this part. The following are the responses to your comments and suggestions:

(1) A number of the previous studies have identified different anthocyanins in berries, and these anthocyanins (both of their content and species) were in high variations among blueberry cultivars. These differences were probably due to their genotypes and growing conditions. However, in the present work, we just focused on the several major anthocyanins detected in the market blueberry extracts, and explored their bio-functions by using in vitro cell experiments.

(2) There were many commercial blueberry extracts on the market, which were the ideal sources for functional food product manufacturing. And many previous studies demonstrated that high doses of anthocyanins showed potential for prevention or treatment of diabetes and its complications. Therefore, it was very necessary to evaluate the content of anthocyanins in commercially available blueberry extracts, and it provided a basis for the follow-up exploration of retinal diseases which related to oxidative stress.

(3) HPLC-Q-TOF-MS/MS results suggested that six different anthocyanins were accurately identified, among which the Cy-3-glu (C3G) was the most abundant, ranging from 376.91 ± 7.91 to 763.70 ± 4.99 μM. The blueberry extract contained higher purity of anthocyanins, and the anthocyanosides reached 342.98 mg/kg. This was much higher than the content of anthocyanins used in many previous large-scale clinical studies (25 or 36 mg/kg in anthocyanins).

However, to follow your suggestions and comments, we made improvements about the part 2.2 of manuscript, and provided extensive explanations and discussions by citing references. (The improved contents were marked in red)

  1. Q2: The antioxidant activity seems a bit confusing, explain in the section 2.3, the comparison of DPPH and FRAP, as both are quite different from each other.

Response: Thank you very much for your comments and suggestions.

By analyzing the total content of phenolics and anthocyanins in the different blueberry extracts, it was suggested that the anthocyanin content in sample 1 was much higher than that in other samples, but there was no significant difference for the phenol content between the sample 1 and the other four samples. In order to explore whether this difference will affect the antioxidant activity of blueberry extracts, we conducted in-vitro cell experiments and evaluated the DPPH scavenging capacity and FRAP capacity. The explanations for these experiment results were as follows:

(1) The antioxidant capacity of the sample S1 which was qualified by the FRAP assay was slightly but significantly lower than other four samples (p< 0.05), moreover, no significantly difference could be observed between the S2, S3, S4 and S5 samples (Fig. 3A(a)). The DPPH scavenging abilities of S1 showed significant difference with S2 and S4 (p< 0.05), but no significant difference with S3 and S5 (Figure 3A(b)). Overall, sample 1 had poor antioxidant capacity compared with the other four samples, which might be due to a fact that it contained less phenolic acids and flavone glycosides compared with other samples. However, this did not mean that S1 sample had no advantage in antioxidant capacity. There was research showed that the antioxidant capacity of blueberries was mainly owing to the contained anthocyanins, which was accounted for as much as 84% of the total an-tioxidant capacity. It can be seen from Fig.3A that the antioxidant capacity of sample 1 was only slightly different from that of other samples.

(2) Because the C3G content of the S1 sample was significantly higher compared with that of the other samples about twice that of the other blueberry extract samples. The antioxidant capacity of C3G was also discussed. It displayed excellent antioxidant capacity.

Therefore, according to your suggestions, for better interpreting these results, we have made improvements about the part 2.3 in the manuscript. The corresponding revised parts are marked in red.

  1. Q3: Insert the following reference as both related to Q-TOF-MS/MS if you find it useful. “10.3389/fphar.2022.890649” and “10.1016/j.jff.2021.104602”.

Response: Thank you very much for your suggestions. Both of these two references are useful, they are good works, therefore, we have added the latter one as cited reference. However, the cannot find the first one, which might be of the wrong doi number. The added reference was marked in red in the manuscript in part 3.3.

  1. Q4: Check the whole manuscript for English as, correct the spelling, table S1 “sequences”. First sentence of third paragraph in introduction, line 58, remove the repeating word. Correct spelling Figure S4, “FRAP”.

Response: Thank you very much for your correction suggestions. We have revised all of the corresponding parts with these spelling mistakes or repeating words. Moreover, in the present manuscript, the FRAP is short for ferric-reducing antioxidant power assay, we have double checked the whole manuscript including the supplementary figures/tables, and revised the wrong spelling FRAP. All the revised contents are marked in red in the manuscript.

Reviewer 2 Report

This article is very interesting because it introduces the importance of foods commonly found in nature as a natural cure for any diseases, such as in this case oxidative stress of the retina due to the anthocyanins found in blueberries. it is important to propose cures using naturally available products that do not interfere with the functions of our bodies (which happens with non-natural medicines). Kudos for this work that explores the chemical parts by associating them with useful health components. For me it can be accepted and published.

Author Response

  1. Comments: This article is very interesting because it introduces the importance of foods commonly found in nature as a natural cure for any diseases, such as in this case oxidative stress of the retina due to the anthocyanins found in blueberries. it is important to propose cures using naturally available products that do not interfere with the functions of our bodies (which happens with non-natural medicines). Kudos for this work that explores the chemical parts by associating them with useful health components. For me it can be accepted and published.

Response: Thank you very much for your recognition of the present work, we also want to express our deep appreciation for your reviewing work. We also believe the present study could provide novel insights in understanding the roles of blueberry anthocyanins for ameliorating oxidative stress induced BRB damage in retina, and paving the way for the applications of this natural berry fruits.

Reviewer 3 Report

The origin of the samples as well as their collection must be specified. The variety of the crop, and how the extracts are obtained. The title is confusing since it talks about marketed products. 6 compounds are identified, but the results are not discussed with the extensive bibliography available, both qualitative and quantitative, from a chemical point of view. 

The following work should be cited and the results discussed. I recommend doing a bibliographic search on blueberry anthocyanins and diabetic retinopathy and including them in the discussion.

Wang C, Wang K, Li P. Blueberry anthocyanins extract attenuated diabetic retinopathy by inhibiting endoplasmic reticulum stress via the miR-182/OGG1 axis. J Pharmacol Sci. 2022;150(1):31-40. doi:10.1016/j.jphs.2022.06.004

Author Response

Firstly, we want to express our deep appreciation for your reviewing work, and we also thank you for your valuable comments and suggestions. According to your comments and suggestions, we have made extensive improvements of the previous manuscript. We have also double checked the languages, grammars and words, and made revisions about the language errors and grammar mistakes. The following are our point by point response to your comments and suggestions.

  1. Q1: The origin of the samples as well as their collection must be specified. The variety of the crop, and how the extracts are obtained. The title is confusing since it talks about marketed products. 6 compounds are identified, but the results are not discussed with the extensive bibliography available, both qualitative and quantitative, from a chemical point of view.

Response: Thank you very much for your valuable comments and suggestions. the following are the responses to your comments and suggestions:

(1) The different blueberries used in the present study came from local varieties in the Great Khingan Mountains (China), where is the major producing area of this fruit, and these samples were harvested in 2021. Blueberry extracts were obtained from Daxinganling Lingonberry Boreal Biotech Co.,Ltd. (Heilongjiang, China), using ethanol extraction method, acquired by multiple extraction. These marketed blueberry extracts products are the good raw materials for producing functional foods or health products in China.

(2) As for the qualitative part about the compounds, By establishing the qualitative and quantitative methods, we have identified six different compounds, however, no more new anthocyanins compounds could be identified, which also might due to the fact that their content are under the detection limit.

Therefore, we just focused on these six different identified anthocyanins compounds, and discussed their bio-functional effects, i.e. antioxidant capacity, according to our experiments results, as well as combined with the previous literatures.

Furthermore, on the one hand, there were few studies on commercially available blueberry extracts, thus, few references can be cited for discussion. On the other hand, the anthocyanin content in blueberries reported in the previous studies was far lower than that in the enriched blueberry extract in the present paper. It is not appropriate for comparing the content or biofunctions effects from different samples by different extraction methods.

However, to follow your suggestions, we have made improvements about the corresponding discussion parts (In part 3.1), where are marked in red in the manuscript.

  1. Q2: The following work should be cited and the results discussed. I recommend doing a bibliographic search on blueberry anthocyanins and diabetic retinopathy and including them in the discussion. Wang C, Wang K, Li P. Blueberry anthocyanins extract attenuated diabetic retinopathy by inhibiting endoplasmic reticulum stress via the miR-182/OGG1 axis. J Pharmacol Sci. 2022;150(1):31-40. doi:10.1016/j.jphs.2022.06.004

Response: Thank you very much for your suggestion, this is a good advice! According to your suggestions, we retrieved and read more literature about the effects of blueberry anthocyanins on diabetes retinopathy. We have included more updated and suitable references in the discussion part, and now, it is a completely new version. All the added contents are marked in red in the manuscript (in the part 2.3 and 2.4).

Reviewer 4 Report

As many studies have reported the identification of anthocyanins from  blueberry fruit,  the author should cleraly pointed out the innovative value of this work. Furthermore, molecular docking (MD) results are insufficient to support the conclusion and more evidence are required. 

Author Response

  1. Q1: As many studies have reported the identification of anthocyanins from blueberry fruit, the author should clearly point out the innovative value of this work. Furthermore, molecular docking (MD) results are insufficient to support the conclusion and more evidence are required.

Response: Thank you very much for your comment! In the results and discussion part, we described the experimental results, and also cited relevant references, and discussed these results, especially the relationship between REDD1protein and anthocyanin monomer compounds. Possible potential mechanisms were also proposed.

In pervious study the effects of anthocyanins from berry fruits on relieving the vision health had been widely explored (Ref: Ashok K. Grover, et al. Mol. Cell Biochem. 2014, 388, 173-183). Moreover, REDD1 played key roles in the DR development and progression, and had widely explored in many previous studies (Ref: Miller, W.P., et al. Free Radic. Biol. Med. 2021, 165, 127-136). Furthermore, baicalein as a natural flavone caused a marked induction of DDIT4 which also called REDD1 that mediated mTOR inhibition and growth inhibition in cancer cells (Ref: Yujun Wang, et al. Cancer Lett. 2015, 368, (2):170-179). Therefore, this study provided a new perspective, namely "Structure identification of the typical blueberry anthocyanins from marketed products and exploring their potential for diabetic retinopathy amelioration". The present study could provide novel insights in understanding the roles of blueberry anthocyanins for ameliorating oxidative stress induced BRB damage in retina.

However, more work should be done in the future to verify these underlying mechanisms using modern molecular analysis methods. In order to follow your suggestion, we carefully checked the contents of the results and discussion section, and improved some sentences to discuss the results from a mechanical perspective.

All added contents are marked in red in the manuscript (in the part 2.5).

Round 2

Reviewer 3 Report

Dear Authors,

The revised version was improved and is ready to be published.

Reviewer 4 Report

The revised version of this paper has addressed most of the problems raised previously, and it is recommended to publish.